# Molecular Epidemic Characteristics and Genetic Evolution of Porcine Circovirus Type 2 in Henan, China

**DOI:** 10.3390/vetsci12040343

**Published:** 2025-04-07

**Authors:** Zhifeng Peng, Huifang Lv, Han Zhang, Li Zhao, Huawei Li, Yanyu He, Kangdi Zhao, Hongxing Qiao, Yuzhen Song, Chuanzhou Bian

**Affiliations:** 1College of Veterinary Medicine, Henan University of Animal Husbandry and Economy, Zhengzhou 450046, China; zfp2017@hnuahe.edu.cn (Z.P.); 80980@hnuahe.edu.cn (H.L.); zhanghan.zz@hotmail.com (H.Z.); 80289@hnuahe.edu.cn (L.Z.); kd18638621887@163.com (K.Z.); 80414@hnuahe.edu.cn (H.Q.); 2Institute of Animal Product Quality and Safety Technology, Henan University of Animal Husbandry and Economy, Zhengzhou 450046, China; 80013@hnuahe.edu.cn; 3The School of Food Technology and Nature Science, Massey University, Palmerston North 4410, New Zealand; y.he@massey.ac.nz; 4College of Marine Sciences, South China Agriculture University, Guangzhou 510642, China

**Keywords:** pigs, porcine circovirus type 2, epidemiology, complete genome, genetic diversity

## Abstract

This study provides an updated overview of the porcine circovirus 2 (PCV2) situation in China following the African swine fever outbreak. The molecular detection rate at population level was 71.17%. The molecular detection rates in 2020, 2021, 2022 and 2023 were 81.16%, 72.41%, 62.50% and 53.49%, respectively. Phylogenetic clustering patterns based on the origin year of the samples were observed, suggesting that nucleotide mutations occur over time. The PCV2d genotype is the dominant genotype in China. The substitution rate for the PCV2 ORF2 sequence in this study was estimated at 1.098 × 10^−3^ substitutions/site/year (ssy), aligning with the high substitution rate expected for PCV2 strains. These findings provide valuable insights into the emergence and evolution of PCV2, and genetic diversity of PCV2, enhancing our understanding of its evolutionary dynamics.

## 1. Introduction

Porcine circovirus-associated diseases (PCVADs) are one of major threats to the swine industry worldwide, causing huge economic losses each year [1,2,3,4,5,6,7]. PCVADs are characterized by immunosuppression and diverse clinical manifestations, including postweaning multisystemic wasting syndrome (PMWS), porcine respiratory disease complex (PRDC), porcine dermatitis and nephropathy syndrome (PDNS), and reproductive disorders in sows [8,9]. These diseases pose serious challenges to the sustainable development of the swine industry worldwide. Key risk factors driving PCV2 transmission include high-density farming practices, suboptimal biosecurity protocols, frequent cross-regional pig movements, and co-infections with pathogens such as PRRSV and *Mycoplasma hyopneumoniae* [10,11,12]. Additionally, incomplete vaccination coverage and viral mutations in immunodominant epitopes further exacerbate viral persistence and spread [11].

The etiologic agent PCV2 is a member of family *Circoviridae*, genus *Circovirus* [13]. To date, nine PCV2 genotypes (PCV2a–PCV2i) have been identified [6]. Epidemiological investigations indicate that prior to 2008, circulating PCV2 strains were predominantly restricted to three genotypes (PCV2a–PCV2c). Among these, PCV2c was initially isolated from asymptomatic porcine sera in Denmark, and later in Brazil, China, and the Netherlands [14,15,16]. PCV2d, identified in 2010, has dominated globally since 2012 [17]. Recent studies further characterize novel genotypes (PCV2e–PCV2i), underscoring ongoing viral diversification [6,18,19]. The genome of PCV2 is predicted to contain 11 open reading frames (ORFs), with ORF1 and ORF2 being the primary functional ORFs [20]. The ORF1 encodes replication-related proteins [21], while the ORF2 encodes the capsid protein (Cap), which facilitates the viral entry and induces the production of neutralizing antibodies [22,23,24]. ORF2 is highly variable and is commonly used in phylogenetic analyses of PCV2 [2,25,26].

PCVADs result in 4–20% mortality in affected herds and reduce average daily weight gain by 15–30% in pigs. Furthermore, in breeding herds, PCV2 infection increases stillbirth rates (≤40%) and reduces litter sizes, with annual losses of GBP 5–10 per sow in the EU [27]. Since its first report in China, PCV2 has become an important pathogen affecting the country’s swine industry [28]. Notably, as the world’s largest pork producer, China incurs annual losses of CNY 12–15 billion (USD 1.7–2.1 billion) due to PCV2-related morbidity. Several studies have investigated the epidemiological and genetic characteristics of PCV2 in different regions of China [3,26,29,30,31,32]. Similar to the global frequency situation, PCV2a, PCV2b, and PCV2d are also the predominate genotypes of PCV2 in China [3,15,33]. Moreover, PCV2 has undergone rapid evolution, with a genotype shift from PCV2b to PCV2d [1,34]. Previous studies have investigated the genetic evolution and genetic characteristics of PCV2 in Henan Province [26,31]. During 2020 and 2021, high pig prices led to the transportation of a large number of piglets to Henan Province. Nevertheless, due to the ongoing epidemic of African swine fever (ASF), most small-scale pig farms were shut down in 2022 and 2023, significantly reducing the density of pig herds and farms. These changes undoubtedly affected the prevalence of PCV2. Notably, while ASF caused acute mortality and immediate production collapse [35], PCV2 imposed chronic economic burdens through subclinical infections and secondary diseases [27]. However, recent data on PCV2 epidemiology remain limited. Given the continuing economic impact of PCV2 on the global swine industry, updating the genetic characteristics of current PCV2 strains is crucial for optimizing PCVAD management and control. In this study, the epidemiological characteristics of PCV2 in Henan Province during 2020 and 2023 were investigated. Additionally, the complete genome sequencing and analysis of 34 PCV2 strains were performed to enhance our understanding of its genetic evolution and diversity.

## 2. Materials and Methods

### 2.1. Sample Collection and Processing

A total of 385 tissue samples were collected from pig farms across 18 cities in Henan Province between February 2020 and May 2023 (clinical case backgrounds summarized in Appendix A). During necropsy, tissue samples including spleen and inguinal lymph nodes were systematically collected from pigs that had died from illness with different clinical symptoms. A subset of samples submitted to the laboratory for analysis also included lung tissue and blood. Samples in this study were from Xinyang (*n* = 16), Zhenghzou (*n* = 11), Shangqiu (*n* = 22), Zhoukou (*n* = 29), Xuchang (*n* = 19), Luoyang (*n* = 18), Kaifeng (*n* = 27), Sanmenxia (*n* = 18), Pingdingshan (*n* = 21), Luohe (*n* = 15), Jiaozuo (*n* = 24), Nanyang (*n* = 26), Zhumadian (*n* = 32), Xinxiang (*n* = 36), Hebi (*n* = 8), Jiyuan (*n* = 9), Puyang (*n* = 16), and Anyang (*n* = 38). Samples were processed following standard operating procedures. The tissue samples from the same pig were diluted with three volumes of phosphate-buffered saline (PBS, pH 7.2), freeze-thawed three times, and homogenized with TissueLyser II (QIAGEN, Germany). Subsequently, the samples were centrifuged at 8000× *g* for 10 min at 4 °C. The supernatants were transferred to a 1.5 mL tube and stored at −80 °C until use. Information on the clinical samples is presented in Figure 1.

### 2.2. DNA Extraction and PCR Detection of PCV2

Viral genomic DNA was extracted from the supernatant with the Magnetic Universal Genomic DNA Kit (TianGen, Beijing, China) following the manufacturer’s instructions. All extracted genomic DNA (310–650 ng/μL) met the minimum threshold for PCR detection and was stored at −80 °C until use. The viral DNA extracted from the tissue samples was used to detect PCV2. To detect PCV2 in clinical samples, a PCR test was performed with the primers (Table 1) as described previously [36]. In brief, the 20 μL reaction mixture consisted of 10 μL of 2× Taq PCR Mix (TianGen, Beijing, China), 0.5 μL (25 μM) of each primer, 1 μL of DNA template, and 8 μL of ddH_2_O. Amplification was performed with an initial denaturation at 94 °C for 10 min, followed by 40 cycles of 94 °C for 30 s, 58 °C for 45 s, 72 °C for 1 min, and a final extension at 72 °C for 10 min. Nucleic acid electrophoresis of PCR products was analyzed by nucleic acid electrophoresis on a 1.5% agarose gel stained with StarStain Red Plus Nucleic Acid Dye (GenStar, Beijing, China) and visualized using ChemiDoc XRS+ Imager (Bio-Rad, Hercules, CA, USA). Samples were considered PCV2-positive if a 486 bp amplicon was detected on the agarose gels using the molecular marker DL2000 DNA ladder (100–2000 bp range; veterinary diagnostic laboratory, Henan University of Animal Husbandry and Economy).

### 2.3. Amplification and Sequencing of Complete Genome of PCV2 Strains

To characterize the PCV2 strains circulating in Henan Province between 2020 and 2023, 1 PCV2-positive sample was randomly selected from each batch of 3–6 samples. A total of 34 complete PCV2 genomes were amplified from the positive samples by PCR with the primers listed in Table 1, as previously described [37]. PCR amplification was performed as follows: initial denaturation at 94 °C for 5 min, followed by 30 cycles of 94 °C for 30 s, 58 °C for 30 s, and 72 °C for 60 s, with a final extension at 72 °C for 10 min. The amplified PCR products were purified using the TIANgel Purification Kit (TianGen, Beijing, China) and subsequently cloned into the vector pMD-18 using the original TA Cloning kit (TaKaRa, Dalian, China). The recombinant plasmids were introduced into *E. coli* DH5α cells via electroporation (2.5 kV, 200 Ω, 25 μF) and cultured on LB agar containing 100 μg/mL ampicillin. To avoid misleading results caused by PCR artefacts, three random clones were sequenced for each of the PCV2 strains. In total, the 102 clones containing PCV2 genomic DNAs were sequenced by Sangon Biotech (Shanghai, China). The full-length PCV2 genomes were assembled using the CExpress software (v6.2, NovelBio Inc., Winnipeg, Canada) with a redundant base coverage strategy (minimum 10 × depth per position). The PCV2 sequences obtained in this study have been deposited in GenBank.

### 2.4. Sequence Comparison and Phylogenetic Analysis

The complete genomes of PCV2 representative strains, available PCV2 commercial vaccine strains, and all PCV2 strains obtained from Henan Province were downloaded from NCBI database (https://www.ncbi.nlm.nih.gov/) (accessed on 28 August 2023) for genetic analysis. The GenBank accession numbers of PCV2 reference strains are listed in Appendix A. Multiple sequence alignments of the PCV2 stains obtained in this study, along with reference strains, were performed using the Clustal W function of the MegAlign V7.1.0 program (Lasergene, DNAStar, Madison, WI, USA). The deduced amino acids of ORF2 were analyzed in a similar manner. Furthermore, phylogenetic reconstruction of the complete PCV2 nucleic acid sequence and ORF2 deduced amino acid sequences was conducted in MEGA 6.0 using the neighbor-joining method under a p-distance model, respectively. The robustness of the topology was evaluated by 1000 bootstrap replications.

### 2.5. Rates of Substitution

The substitution rates of the PCV2 ORF2 nucleotide sequences were estimated using the Bayesian Markov chain Monte Carlo (MCMC) algorithm implemented in BEAST v1.10.4 to [38]. Three independent MCMC computations were performed on 200 PCV2 ORF2 sequences under the relaxed molecular clock model, the Hasegawa–Kishono–Yano (HKY) substitution model, and the γ-site heterogeneity model. To account for dynamic population changes over time, the Bayesian Skygrid tree was selected, configured with 52 parameters and a single transition point at the end [39]. MCMC chains were run for 2 × 10^8^ iterations, with posterior probability distributions sampled every 1000 steps. Convergence was assessed using Tracer v1.7.2 via visual inspection of trace plots and ensuring an effective sample size exceeded 200 after a 10% burn-in. Substitution rates from three independent runs were obtained using LogCombiner v1.10.4 (part of the BEAST v1.10.4 package), with 95% confidence intervals calculated. Subsequently, the MCMC runs were independently repeated on subsets of the PCV2a, PCV2b, and PCV2d cap gene nucleotide sequence data using the same methodology.

## 3. Results

### 3.1. The Frequency of PCV2 in Different Regions During 2020 and 2023

From 2020 and 2023, a total of 385 clinical samples were collected, with 274 testing positive for PCV2, yielding an overall positivity rate of 71.17% (274/385) (Table 2). Among the 18 regions in Henan Province, the top three detection rates of PCV2 were 84.38% in Zhumadian, 79.31% in Zhoukou, and 78.95% in Anyang. The lowest detection rate of PCV2 was 55.56% in Sanmenxia (Figure 1). A temporal analysis of PCV2 detection rates, conducted using SPSS v28.0 with sampling duration-adjusted weights, revealed a significant decreasing trend over time: 81.16% (112/138) in 2020, 72.41% (84/116) in 2021, 62.50% (55/88) in 2022, and 53.49% (23/43) in 2023 (χ^2^ = 9.8, df = 3, *p* = 0.021). Post hoc comparisons with Bonferroni correction confirmed a marked decline between 2020 and 2023 (*p* = 0.003). However, key limitations should be noted. The small sample size (*n* = 43) may have reduced statistical power, and potential residual bias may exist due to the truncated 2023 sampling period (five months). Additionally, weighting adjustments for partial-year data introduce inherent assumptions that require careful interpretation. Despite these constraints, the observed decline suggests a potential epidemiological shift in PCV2 prevalence, warranting further validation through standardized longitudinal surveillance programs.

### 3.2. Genome Sequence Analysis of PCV2

To characterize the genetic diversity of recently prevalent PCV2 strains in Henan Province, 34 PCV2-positive samples were randomly selected from multiple regions for full-genome amplification, sequencing, and analysis. All 34 PCV2 strains had a genome length of 1767 nt. The following sequences were deposited in GenBank: OQ749408-OQ749433, OR114683-OR114690 (Table 3). The ORF1 gene and ORF2 genes were 945 nt and 702 nt in length, respectively. Pairwise sequence comparisons revealed nucleotide homologies ranging from 94.10% to 100.00% for the complete genome, 96.60% to 100.00% for ORF1, and 89.00% to 100.00% for ORF2. At the amino acid level, homology was found to be 98.10% to 100.00% for Rep and 87.20% to 100.00% for Cap.

### 3.3. Phylogenetic Analysis of PCV2 Strains

Phylogenetic analysis of PCV2 strains based on complete PCV2 genome sequences and ORF2 amino acid sequences from the 34 strains showed that three sub-genotypes (PCV2a, PCV2b, and PCV2d) were identified in the current study (Figure 2), 58.82% (20/34) strains belonged to the sub-genotype PCV2d; and 35.30% (12/34) strains belonged to the sub-genotype PCV2b. Only two strains (5.88%, 2/34), HNAY-1-2020 and HNSMX-2023, were clustered with PCV2a strains. The percentages of PCV2d strains in 2021 (87.50%), 2022 (62.50%) and 2023 (75.00%) were higher than that in 2020 (20.00%) (Figure 3). Noticeably, 90.00% (18/20) of PCV2d strains were collected during 2021–2023, whereas 41.67% (5/12) PCV2b strains were collected during the same period.

### 3.4. Mutation Analysis of Cap Proteins

To explore the characteristics of the amino acid sequences of Cap of the PCV2 strains in this study, the amino acid sequence alignment was performed between the 34 PCV2 strains and the 28 reference PCV2 strains, including PCV2 commercial vaccine strains, LG (HM038034.1, PCV2a), WuHan (FJ5980441.1, PCV2b), and DBN-SX07 (HM641752.1, PCV2b) (Appendix A). The results showed that the lengths of Cap amino acid sequences of 34 PCV2 strains were 234 amino acids. Genotype-specific substitutions were identified: PCV2b exhibited V57I, I89R/L, P134T, and D210E (absent in PCV2a/d), while PCV2d showed S68N, I89L, T/P134N, and S169G/R (absent in PCV2a/b). In addition, in this study, no amino acid deletions or insertions were observed among the 34 PCV2 strains. Thirty-one critical amino acid substitution sites were found in the Cap proteins, predominantly clustering in immunodominant regions (Table 4).

### 3.5. Rates of Substitution

Bayesian MCMC analysis of the PCV2 ORF2 nucleotide sequences revealed evolutionary rates, which are summarized in Table 5. Convergence diagnostics confirmed stable posterior distributions across three independent runs (10% burn-in), with effective sample size (ESS) values exceeding 200 for all parameters. Genotype-specific analyses demonstrated distinct substitution rates: PCV2a (4.658 × 10^−4^ ssy), PCV2b (6.476 × 10^−4^ ssy), and PCV2d (2.127 × 10^−3^ ssy). When all three genotypes were analyzed collectively, the substitution rate for the PCV2 ORF2 nucleotide sequences was estimated to be 1.098 × 10^−3^ ssy.

## 4. Discussion

PCV2 has been endemic in Chinese swine populations for decades, with PCVADs posing significant economic and health challenges to the swine industry [1,34,40]. The rapid evolution of PCV2 is driven by international breeding pig transportation, natural viral evolution, and immune pressure from vaccination [36]. Two major genotype transitions have been documented in China: PCV2a dominated prior to 2003, PCV2b emerged as the primary genotype between 2003 and 2010, and PCV2d has become predominant since 2010 [11,36,41,42]. Notably, the co-circulation of multiple genotypes within individual farms highlights the genetic plasticity of PCV2 and complicates disease management [1,30,43,44].

In this study, PCV2 was detected in 71.17% (274/385) of tissue samples collected from Henan Province (2020–2023), a rate comparable to historical regional data (62.4–72.9%, 2015–2019) [30,31], but higher than that reported in Shanghai (57.78%) [2], Yunnan (60.93%) [45], Shandong (36.98%) [11], and Hebei Province (39.96%) [32]. Taken together, these aforementioned studies indicated a high PCV2 detection rate in China. The consistently high detection rates suggest widespread PCV2 circulation in China. However, our findings indicate a gradual decline in detection rates from 2020 to 2023, likely due to structural transformations within the swine industry. Henan Province dominated China’s swine production, with tens of millions of pigs raised in thousands of pig farms of various scale every year. Owing to the high pig price (from 2020 to 2021), millions of piglets were transferred to Henan Province, which may be the reason of high frequency of PCV2 from 2020 to 2021 in Henan Province. Nevertheless, post-ASF industry restructuring led to the closure of small-scale farms with poor biosecurity, reducing swine density and inter-regional transport—key drivers of PCV2 transmission—which may explain the declining detection rates from 2022 onward.

To date, phylogenetic analyses of PCV2 genomic sequences currently delineate nine genotypes (PCV2a–i), with China exhibiting sustained co-circulation of multiple genotypes [2,34,46]. Notably, evolutionary surveillance since 2020 has identified PCV2d as the dominant circulating genotype in Chinese swine populations, replacing previously prevalent strains [2]. In this study, phylogenetic trees based on whole genome sequences and ORF2 encoded amino acid sequences classified the 34 PCV2 strains PCV2a (5.88%, 2/34), PCV2b (35.30%, 12/34), and PCV2d (58.82%, 20/34), reinforcing PCV2d’s predominance in Henan Province. This finding aligns with previous studies, including a 2018–2020 central China survey (PCV2d: 47.06%) [26] and a 2020–2021 Henan-specific analysis (PCV2d: 40.00%) [47], confirming the ongoing shift toward PCV2d dominance in China’s swine populations.

The capsid (Cap) protein, the sole structural protein of PCV2, plays a crucial role in viral entry, replication, and host immune responses [46,48]. Specially, residues 1–41 are involved in the nuclear localization of the virus [49], while residues 47–63 are essential for the recognition of PCV2 epitopes [50]. Notably, PCV2 exhibits higher substitution rates (10^−3^–10^−4^ substitutions/site/year) than typical DNA viruses [51,52], as evidenced by genotype-specific rates in this study: PCV2a (4.658 × 10^−4^ ssy), PCV2b (6.476 × 10^−4^ ssy), and PCV2d (2.127 × 10^−3^ ssy). The accelerated evolution of PCV2 likely contributes to amino acid substitutions in Cap, potentially altering viral characteristics and immune escape potential [23,33].

Analysis of ORF2 sequences from the 34 PCV2 strains revealed substantial variability (nucleotide: 88.90–100.00%; amino acid: 88.60–100.00%). In addition, three variable regions (residues 53–91, 121–136, 190–206) (Table 4) overlapped immunodominant domains (residues 65–87, 113–147, and 193–207) in the Cap protein [53]. Notably, residue 59 (Alanine in PCV2a), a neutralizing epitope, exhibited mutations that may facilitate antigenic drift and variant emergence [54]. While previous studies have postulated a correlation between Cap protein variability and PCV2 pathogenicity [54], virulence determinants are likely multigenic. Therefore, comprehensive structural analyses of genotype-specific antigenic conformations are needed to elucidate pathogenesis mechanisms and guide the development of high-efficacy PCV2 vaccines.

This study has several limitations: first, its small sample size (*n* = 34) and geographic biases; second, temporal sampling discontinuities hindering evolutionary trend resolution. Despite these constraints, our findings provide insights into PCV2 evolution and epidemiology. Future studies should address these limitations by incorporating larger cohorts and longitudinal sampling strategies.

## 5. Conclusions

This study delineates critical epidemiological transitions in PCV2 within one of China’s major swine-producing regions (Henan Province), where declining detection rates and genotype shifts (PCV2d dominance) mirror global trends, providing actionable insights for refining vaccine strategies and containment protocols in intensive pig production systems worldwide. Observed antigenic divergence in PCV2d Cap epitopes highlights potential gaps in PCV2a-based vaccine efficacy, urgently requiring PCV2d-targeted or multivalent vaccine development. Despite post-African swine fever (ASF) biosecurity improvements, persistent co-circulation of multiple genotypes highlights the necessity to integrate real-time genomic surveillance with enhanced containment measures, including optimized farm biosecurity protocols and regionally tailored vaccination programs. Collectively, these measures are critical to mitigate economic losses and preempt PCVAD resurgence in China’s swine industry.

## Figures and Tables

**Figure 1 vetsci-12-00343-f001:**
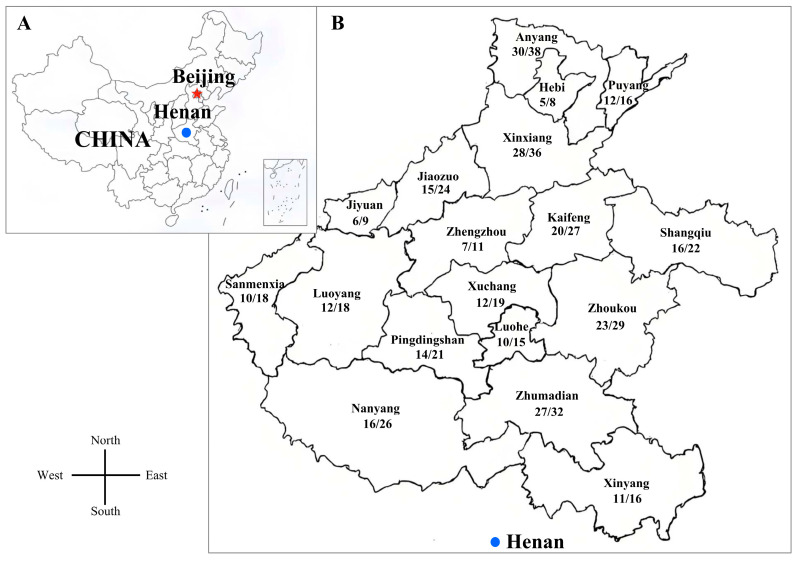
The geographical locations in Henan Province of the samples collected in this study. (**A**) Geographic location of Henan Province in China. (**B**) The geographical locations in Henan Province of China where 385 samples were collected. The numbers indicate the detection rate of PCV2 in different cities. Red star: Beijing (China’s capital); Blue dot: Henan Province.

**Figure 2 vetsci-12-00343-f002:**
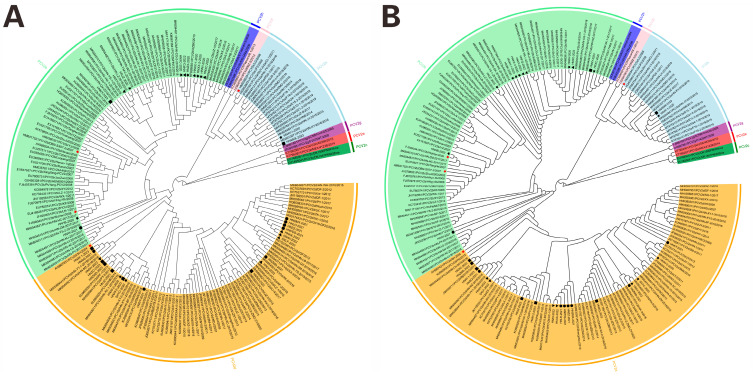
Phylogenetic trees based on the PCV2 strains obtained in this study and reference strains. (**A**) Phylogenetic tree of the complete genome sequences. (**B**) Phylogenetic tree of deduced amino acid sequences of ORF2. The phylogenetic reconstruction was performed in MEGA6.0 software using the neighbor-joining algorithm with 1000 bootstrap replicates. Red star represented PCV2 commercial vaccine strains. Black stars, black circles, black triangles and black squares represented PCV2 strains obtained in this study in 2020, 2021, 2022 and 2023, respectively.

**Figure 3 vetsci-12-00343-f003:**
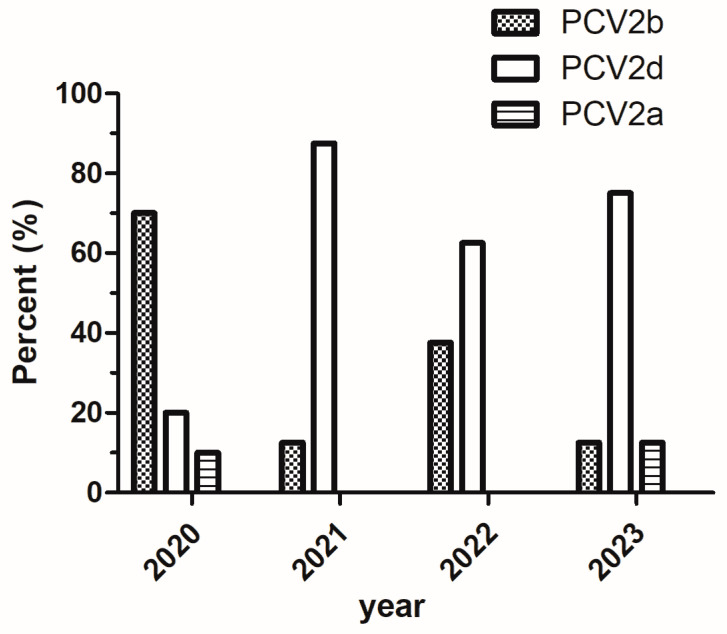
The proportion of PCV2 subtypes in different years in this study. A total of 385 clinical samples were collected from pig farms in 18 cities in Henan Province during February 2020 and May 2023, with 274 testing positive for PCV2. The percentages of PCV2d strains in 2021 (87.50%), 2022 (62.50%) and 2023 (75.00%) were higher than that in 2020 (20.00%).

**Table 1 vetsci-12-00343-t001:** Primers used in this study.

Primer Name	Sequence (5′−3′)	Length	Annealing Temperature	Purpose
PCV2-F	CTGTTTTCGAACGCAGTGCC	486 bp	56 °C	Detection of PCV2 [36]
PCV2-R	GCATCTTCAACACCCGCCT			
PCV2-C-F	CGGGGTACCACTGAGTCTTTTTTATCACTTCG	1767–1768 bp	58 °C	PCV2 whole-genome amplification [37]
PCV2-C-R	CCCAAGCTTAAGACTCAGTAATTTATTTCATATGG			

**Table 2 vetsci-12-00343-t002:** The frequency of PCV2 in different regions in Henan Province of China during 2020 and 2023.

Region	No. of Tested Samples	No. of PCV2 Positive Samples	Frequency
Anyang	38	30	78.95%
Puyang	16	12	75.00%
Hebi	8	5	62.50%
Xinxiang	36	28	77.78%
Jiaozuo	24	15	62.50%
Jiyuan	9	6	66.67%
Zhenzhou	11	7	63.64%
Kaifeng	27	20	74.07%
Shangqiu	22	16	72.73%
Zhoukou	29	23	79.31%
Xuchang	19	12	63.16%
Luohe	15	10	66.67%
Zhumadian	32	27	84.38%
Pingdingshan	21	14	66.67%
Nanyang	26	16	61.54%
Luoyang	18	12	66.67%
Sanmenxia	18	10	55.56%
Xinyang	16	11	68.75%
Total	385	274	71.17%

**Table 3 vetsci-12-00343-t003:** Detailed information of PCV2 strains sequenced in this study.

Strain	Collection Year	Isolation Origin	Genotype	Accession No.	Length (nt)
HNZK-2020	2020	Zhoukou, Henan	PCV2b	OQ749408	1767
HNXX-1-2020	2020	Xinxiang, Henan	PCV2b	OQ749409	1767
HNNY-2020	2020	Nanyang, Henan	PCV2b	OQ749410	1767
HNJZ-2020	2020	Jiaozuo, Henan	PCV2b	OQ749411	1767
HNXC-2020	2020	Xuchang, Henan	PCV2b	OQ749412	1767
HNAY-1-2020	2020	Anyang, Henan	PCV2a	OQ749413	1767
HNXX-2-2020	2020	Xinxiang, Henan	PCV2b	OQ749414	1767
HNHB-2020	2020	Hebi, Henan	PCV2b	OQ749415	1767
HNAY-2-2020	2020	Anyang, Henan	PCV2d	OQ749416	1767
HNJY-2020	2020	Jiyuan, Henan	PCV2d	OQ749417	1767
HNAY-2021	2021	Anyang, Henan	PCV2b	OQ749418	1767
HNZZ-2021	2021	Zhengzhou, Henan	PCV2d	OQ749419	1767
HNPDS-2021	2021	Pingdingshan, Henan	PCV2d	OQ749420	1767
HNSQ-2021	2021	Shangqiu, Henan	PCV2d	OQ749421	1767
HNLH-2021	2021	Luohe, Henan	PCV2d	OQ749422	1767
HNNY-2021	2021	Nanyang, Henan	PCV2d	OQ749423	1767
HNZMD-2021	2021	Zhumadian, Henan	PCV2d	OQ749424	1767
HNSMX-2021	2021	Sanmenxia, Henan	PCV2d	OQ749425	1767
HNXX-2022	2022	Xinxiang, Henan	PCV2d	OQ749426	1767
HNAY-2022	2022	Anyang, Henan	PCV2d	OQ749427	1767
HNXY-2022	2022	Xinyang, Henan	PCV2b	OQ749428	1767
HNXC-2022	2022	Xuchang, Henan	PCV2b	OQ749429	1767
HNLY-2022	2022	Luoyang, Henan	PCV2d	OQ749430	1767
HNNY-2022	2022	Nanyang, Henan	PCV2d	OQ749431	1767
HNFQ-2022	2022	Xinxiang, Henan	PCV2b	OQ749432	1767
HNLY-2022	2022	Luoyang, Henan	PCV2d	OQ749433	1767
HNJZ-2023	2023	Jiaozuo, Henan	PCV2d	OR114683	1767
HNZMD-2023	2023	Zhumadian, Henan	PCV2d	OR114684	1767
HNXC-2023	2023	Xuchang, Henan	PCV2d	OR114685	1767
HNSQ-2023	2023	Shangqiu, Henan	PCV2d	OR114686	1767
HNNY-2023	2023	Nanyang, Henan	PCV2b	OR114687	1767
HNAY-1-2023	2023	Anyang, Henan	PCV2d	OR114688	1767
HNSMX-2023	2023	Sanmenxia, Henan	PCV2a	OR114689	1767
HNAY-2-2023	2023	Anyang, Henan	PCV2d	OR114690	1767

**Table 4 vetsci-12-00343-t004:** Critical amino acid substitutions in the Cap proteins of PCV2 strains identified in this study.

Position	PCV2a	PCV2b	PCV2d	Position	PCV2a	PCV2b	PCV2d
8	F	F_(1/12_/Y_(11/12)_	F_(16/20)_/Y_(4/20)_	121	S	S	T
47	A	T	T	131	M	T	T
53	F	F_(11/12)_/I_(1/12)_	I	133	V	A	A
57	V	I	V	134	P	T	N
59	A	K_(11/12)_/R_(1/12)_	K	136	Q	L	L
63	S	K_(1/12)_/R_(11/12)_	R	151	P	T	T
68	S	A	N	169	S	S_(11/12)_/C_(1/12)_	R_(3/20)_/G_(17/20)_
73	L	M	M	185	M	L	L
77	L	I	I	190	S	A_(1/12)_/T_(11/12)_	T
78	D	N	N	191	R	G	G
80	V	L	L	206	T/K	I	I
86	T	S	S	210	D	E	D
88	K	P	P	215	V	V	I
89	I	R_(11/12)_/L_(1/12)_	L	232	N	N	N
90	S	S	T	234	K	K_(1/12)_/_(11/12)_	K
91	I	V	V				

**Table 5 vetsci-12-00343-t005:** Substitution rates of PCV2 ORF2 nucleotide sequences.

PCV2 Genotype	Mean Rate of Substitution (Substitution/site/year)	95% Highest Posterior Density	Effective Sample Size
PCV2a	4.658 × 10^−4^	8.643 × 10^−4^	245,643
PCV2b	6.476 × 10^−4^	1.063 × 10^−3^	254,638
PCV2d	2.127 × 10^−3^	2.853 × 10^−3^	67,627
Combined analysis for all genotypes	1.098 × 10^−3^	1.381 × 10^−3^	81,829

## Data Availability

All data generated or analyzed during this study are included in this published article [and its Appendix A].

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
