# Peer review of "Molecular Epidemic Characteristics and Genetic Evolution of Porcine Circovirus Type 2 in Henan, China"

_vetsci, 2025, doi:10.3390/vetsci12040343_

Round 1

Reviewer 1 Report

Comments and Suggestions for Authors

Comments for the authors

Abstract

  • In the keywords place also pigs, it is the focus species of the study.
  • It could be more concise and focus on key findings, such as the dominance of PCV2d and the high substitution rate.

Introduction

  • Give a more detailed background on the PCV2 genotypes.
  • Provide more detail on the huge economic impact of PCV2 on swine industry worldwide so that you can support the importance of continued surveillance of the virus to successfully combat it

Materials and Methods

  • Please provide details of tissue sampling. Why were spleen and inguinal lymph nodes were selected?
  • Were the pigs included in the study previously vaccinated against PCV2?
  • Describe the criteria for selecting the 34 PCV2 strains for whole genome sequencing.

Results

  • A table summarizing detection rates by year and region could be beneficial.

Discussion:

  • You could discuss some potential drawbacks of your study.
  • You could emphasize the need for continuous monitoring.
Comments on the Quality of English Language
  • Moderate editing regarding English language is required for the improvement of the text.

Reviewer 2 Report

Comments and Suggestions for Authors

The present study describes the situation of Porcine Circovirus type 2 in Henan Province, China. It diagnoses samples collected from 2020 to 2023 in various cities using PCR, sequences the complete genome of several strains, determines phylogenetic trees, and analyzes amino acid substitutions in the cap gene. Below are the suggestions from this reviewer:
-Lines 70-71:Remove from the introduction; it could be placed in the conclusion.  
-Lines 94-96:Verify the description of the cycles used.  
-Line 121:In Supplementary Table 2, are the 18 sequences from the last group from 2020? It is necessary to specify the year.  
-Line 205:Rewrite, as the percentages refer to all three years, not only 2021 and 2023; the same mistake appears in line 222.  
-Lines 206-207:The citation for Figure 2 should appear after the previous sentence, i.e., after "(20.00%)" and before the period, as it refers to this data.  
-The phylogenetic trees are illegible in the revised PDF version; could they be added to the supplementary material in a higher-resolution version?  
-Lines 240-242:Remove, as they are not a table "note" and are already included in the discussion.  
-Discussion:Needs restructuring. Multiple discussion points are presented within single paragraphs, while some points continue from one paragraph to another. As in the rest of the manuscript, the English needs improvement. Some sections are difficult to understand due to poor English, and there are also repeated words. 

Comments on the Quality of English Language

The English needs improvement. Some sections are difficult to understand due to poor language quality.

Reviewer 3 Report

Comments and Suggestions for Authors

Line 41: It is suggested to include economic figures to enhance the impact of the text.

Line 46: It is recommended to add the main risk factors in the spread of this virus.

Line 53: Before addressing the impact in China, it would be advisable to include information on the global impact of circovirus, particularly in countries with significant relevance in pig export and import.

Line 61: A brief comparison between the losses caused by African Swine Fever and circovirus is suggested to broaden the perspective of the study's impact.

Line 77:

  • Was a specific criterion established for the selection of animals, such as growth stage, age, weight, or any other characteristic that grouped them?
  • Was a prior suspicion of circovirus considered, or was the sanitary status of the location, including vaccination schemes, taken into account?

Line 89: It is recommended to include data on the concentration and quality of the obtained DNA.

Line 99: It is suggested to include information on the molecular marker used and its corresponding scale, as well as the laboratory where the analysis was conducted (in parentheses).

Line 101: Why were no other molecular variants of PCV2 explored? Was the use of primers limited to the PCV2-C variant only?

Line 104:

  • The text should specify the criteria under which the samples were selected.
  • Was a random criterion used, the highest viral load, or another epidemiological factor?
  • The number of samples per batch should be clarified in the text.

Line 106: Was the entire genome amplified using only these primers? This is not entirely clear.

Line 110: Was any antibiotic used in the culture medium for the cloning process? If so, it is suggested that the author add this information.

Line 112: It is recommended to specify the transformation method used (heat shock, electroporation, etc.).

Line 116:

  • It is necessary to add information about the assembly strategy used.
  • Was specific software employed for the analysis?

Line 147: The epidemiological results presented are very interesting. However, it is necessary to clarify whether there are statistically significant differences. Without statistical validation, it cannot be concluded whether the changes in PCV2 positivity are real or due to chance.

Line 159: Were analyses performed to adjust for sample size bias? It is important to clarify this in the text so that readers can correctly interpret the results, considering that the proportion of samples per year is not uniform.

Line 160: Since the study has epidemiological significance, it is suggested to complement the table with a map showing the mentioned regions, indicating in each the metrics of positive samples for PCV2. This would allow readers to visually understand the virus's dynamics in the study region.

Line 198: Was the selection criterion for these 140 samples based solely on their origin from the same region, or were specific genomic or pathogenic characteristics also considered?

Line 210: Figure 1 is not clearly visible. It is recommended to improve its quality to facilitate interpretation by the reader. If this is the final version, it should be optimized before publication.

Line 219: This information does not align with the methodology. It is recommended to clarify why identification results for PCV2b, PCV2d, and PCV2a are presented if it was initially stated that the primers used were specific only for PCV2-C.

Line 332: It is suggested to include a statement on the global impact of the study’s findings rather than limiting it solely to the analyzed region.

Line 337: From a health perspective, do these findings provide insight into whether PCV2a-based vaccines remain effective, or is an update needed? If so, including this opinion could enrich the conclusion.

Line 338: Could the results suggest that current control strategies need adjustment? If so, it is recommended to introduce this perspective in the conclusion as well.

Round 2

Reviewer 1 Report

Comments and Suggestions for Authors

The authors have thoroughly addressed all of my comments and concerns and have provided clear and well-reasoned responses that enhance the clarity and overall quality of their manuscript. They have made the necessary revisions and included additional explanations and methodological clarifications where needed. The revised version resolves any ambiguities and strengthens the manuscript’s contributions to the field. Given these improvements, I am satisfied with their responses and revisions and believe that the manuscript is now suitable for publication.